# Probabilistic Bisection Algorithm Provably Achieves Exponential Convergence

Ganghua Wang [1]   Yuwei Cheng [2]   Haifeng Xu [1 3]

## Abstract

The probabilistic bisection algorithm (PBA) extends the classical binary search to settings with noisy responses, and is a foundational algorithm commonly used in basic problems such as root-finding. Despite its strong empirical success, its theoretical property, particularly the convergence rate, remains unclear. This paper establishes that PBA converges at a geometric rate, providing a rigorous justification for its empirical efficiency. Notably, this rate is optimal in the sense that it matches the performance of classical binary search under noiseless responses. The core of our analysis lies in directly characterizing the dynamics of PBA queries, which had not been examined in the prior literature. We show that the queries oscillate around the truth but steadily draw closer, thus leading to an estimator that rapidly concentrates on the truth. Beyond resolving the long-standing question of PBA's convergence, our developed techniques offer new tools for analyzing PBA's dynamics, which may be of independent interest.

## 1. Introduction

Binary search is a fundamental algorithm that addresses the core challenge of **efficiently locating a target within an ordered space** using the principle of divide-and-conquer. It underpins a wide range of modern algorithms in computer science, statistics, and applied mathematics (Knuth, 1997; Karp & Kleinberg, 2007; Waeber et al., 2013), and serves as a building block for systems and methods from multidimensional data to search on graphs and trees (Bentley, 1975; Nowak, 2009; Emamjomeh-Zadeh et al., 2016; Rodriguez & Ludkovski, 2020a). Classical applications include fast key

---
[1]Data Science Institute, University of Chicago, Chicago, IL, USA [2]Department of Statistics, University of Chicago, Chicago, IL, USA [3]Department of Computer Science, University of Chicago, Chicago, IL, USA. Correspondence to: Ganghua Wang <ganghua@uchicago.edu>.

*Proceedings of the 43rd International Conference on Machine Learning*, Seoul, South Korea. PMLR 306, 2026. Copyright 2026 by the author(s).

retrieval in large databases and numerical root-finding in engineering and economics. For instance, consider finding a unique root of a monotone function $h : [0, 1] \to \mathbb{R}$ where one can query only the sign of $h(x)$. When the response to each query $x$ is noiseless, a binary search algorithm efficiently locates the root by halving the search interval each round. After $n$ queries, the remaining interval has length $2^{-n}$, achieving the optimal exponential convergence rate.

In practice, however, the observed responses are often noisy, e.g., due to transmission and measurement error, meaning that they have a chance to be incorrect. Motivated by noisy channel coding, the Probabilistic Bisection Algorithm (PBA, Horstein, 1963) extends binary search to handle noisy labels. Compared to binary search, PBA adopts a Bayesian approach to select the query. In the 1-D root-finding setup, PBA maintains a probability distribution with density $f_t$ over the support $[0, 1]$, representing the likelihood of each point being the true root. At each round $t$, PBA queries the median $x_t$ of this distribution and receives a noisy response $y_t$ indicating the sign at $x_t$. The belief is then updated via Bayes' rule given $y_t$. For instance, if $y_t$ is positive, then $f_t(x) = 2(1 - p)f_{t-1}(x)$ for $x \le x_t$ and $f_t(x) = 2pf_{t-1}(x)$ for $x > x_t$, where $p$ is the noise level. The process repeats until termination, with the final estimator being the last query.

Despite strong empirical performance, (Waeber, 2013; Frazier et al., 2019; Rodriguez & Ludkovski, 2020a;b), PBA's theoretical property, particularly its convergence rate, remains poorly understood. The difficulty stems from its **intricate query process over a continuous search space**. In comparison, the so-called noisy binary search typically focuses on a finite search set and enjoys well-understood guarantees. It has been shown that locating a target among $H$ elements with error probability at most $\delta$ requires only $O(\log(H/\delta))$ queries (Karp & Kleinberg, 2007; Nowak, 2009; Emamjomeh-Zadeh et al., 2016). However, these results rely crucially on the discretized structure of the search space *such as searching nodes on a path (Aslam & Dhagat, 1991; Karp & Kleinberg, 2007) or a graph (Emamjomeh-Zadeh et al., 2016; Dereniowski et al., 2019)*. In contrast, PBA addresses a continuous domain with uncountably many possible queries, which requires fundamentally different analysis tools.

Historically, analyzing PBA's performance has been difficult due to the continuous nature of the query sequence, which makes tracking the estimation non-trivial. As a result, prior efforts either (1) adopted a discretized version of PBA, or (2) invoked a Bayesian framework where the unknown truth is modeled as a continuous random variable. **However, these approaches are unable to directly characterize the convergence behavior of the original PBA given a fixed truth**. Specifically, Burnashev & Zigangirov (1974) proposed a discretized version of PBA, which we refer to as the BZ algorithm. BZ restricts queries to a finite grid $\{0, 1/K, 2/K, \dots, 1\}$ for some constant $K$. With carefully modified update and query rules, they proved that BZ attains exponential convergence when $K$ adapts to the query size (Burnashev & Zigangirov, 1974; Castro & Nowak, 2008). However, a pre-selected and fixed $K$ is required to run BZ in practice, thus such a convergence rate cannot be expected. Waeber et al. (2013) analyzed PBA in a Bayesian setting. By modeling the root as a random variable $X^*$ uniformly distributed on $[0, 1]$, they proved that $\mathbb{E}|X^* - \widehat{X}_n|$ decays geometrically, where $\widehat{X}_n$ is the PBA estimate after $n$ queries. However, this result hinges critically on the assumption that $X^*$ is a continuous random variable. Therefore, this analysis does not apply to real-world tasks where the ground truth is a fixed but unknown constant, such as root-finding and boundary detection problems.

A closer inspection of these approaches shows that the main barrier to analyzing the original PBA, again, lies in the complex, location-dependent behavior of its queries. Both the discretized analysis of Burnashev & Zigangirov (1974) and the Bayesian analysis of Waeber et al. (2013) exploit a simplifying property: at every round a quantity that upper-bounds the estimation error is expected to decrease, regardless of where the queries fall. Unfortunately, this guarantee breaks down when analyzing the original PBA with a fixed ground truth, as the improvement in accuracy depends delicately on the query locations (with further discussion in Subsection 2.2).

Our work demonstrates that understanding the query behavior of PBA is both essential and powerful in tackling this problem. In Subsection 2.3, we develop new analytical techniques that measure the improvement contributed by the query at each round, and characterize the number of queries that lead to a better estimation, an aspect not studied in the prior literature. These tools allow us to directly study the dynamics of PBA queries. Intuitively, we show that the queries oscillate around the ground truth but steadily draw closer, driving the posterior distribution to concentrate sharply at the ground truth.

Building on these tools, we prove that **PBA converges at an exponential rate for any fixed, unknown ground truth.** The rate we establish is optimal, matching the ge-

ometric convergence achievable by classical binary search with noiseless feedback. This result settles the long-standing theoretical question of whether PBA retains its empirical efficiency under noisy responses (Waeber et al., 2013). Moreover, our developed tools provide a fine-grained understanding of PBA's query process, which may be of independent interest for other adaptive algorithms.

The rest of the paper is organized as follows. Section 2 present our main result, the convergence rate of PBA for one-dimensional data. Simulation experiments are conducted in Section 4, and we discuss the extension to the high-dimensional data in Section 3. We conclude the paper with further discussions in Section 5.

## Conflict of Interest Disclosure

The authors declare no conflict of interest.

## 2. Convergence Rate of PBA

### 2.1. Setup

We cast the root-finding problem as a special case of binary classification. Consider a learner seeking to identify the unknown classifier $h_{\theta^*}(x) = \mathbb{1}_{x \geq \theta^*}$ within a hypothesis class $\mathcal{H} = \{h_\theta : \theta \in [0, 1]\}$, where $\mathbb{1}_{(\cdot)}$ is an indicator function. Let $p \in (0, 1/2)$ denote the noise level in the response. In this formulation, $\theta^*$ is the unknown root, and each response is flipped independently with probability $p$. Specifically, for any query $X$, the observed response $Y$ satisfies that $\mathbb{P}(Y = h_{\theta^*}(X)) = 1 - p$ and $\mathbb{P}(Y = 1 - h_{\theta^*}(X)) = p$. We note that our results also extend to the more general setting where $\mathbb{P}(Y = 1 - h_{\theta^*}(X)) \leq p$, as elaborated in Appendix B.

**Probabilistic Bisection Algorithm (PBA).** A learner can use PBA to efficiently estimate $\theta^*$ as follows. Let $P_0$ be a uniform prior distribution such that its density function is $f_0(x) = 1, x \in [0, 1]$. At round $i \geq 1$, PBA will select a query $X_i$ as the median of $\mathbb{P}_{i-1}$, i.e.,

$$\mathbb{P}_{i-1}(X \leq X_i) = 1/2.$$

After observing the corresponding label $Y_i$, PBA updates the posterior distribution as follows:

(1) If $Y_i = 1$, $f_i(x) = \begin{cases} 2(1-p)f_{i-1}(x), & x \leq X_i, \\ 2p f_{i-1}(x), & x > X_i, \end{cases}$

(2) If $Y_i = 0$, $f_i(x) = \begin{cases} 2p f_{i-1}(x), & x \leq X_i, \\ 2(1-p)f_{i-1}(x), & x > X_i. \end{cases}$

The posterior distribution at round $i$ is $\mathbb{P}_i(t) = \int_0^t f_i(x)dx$. The final estimator of $\theta^*$ after $n$ rounds is $\widehat{\theta}_n := X_{n+1}$.

*Remark* 1 (Prior and Posterior Distributions). In our setting

the unknown root $\theta^*$ is fixed. The distribution $\mathbb{P}_i$ represents the learner's belief about $\theta^*$: at round $i$, they believe that the probability that $\theta^* \leq t$ is given by $\mathbb{P}_i(t)$. We use the terms *prior* and *posterior* in keeping with the PBA literature, where the algorithm is commonly interpreted from a Bayesian perspective.

## 2.2. Exponential Convergence Rate

Throughout the paper, $c$ and $C$ are either universal constants or constant of $p$ only, though their value may vary from line to line. We use the terms 'root', 'truth', and 'ground truth' interchangeably. The complete proof of Theorem 1 is included in Subsection A.1.

**Theorem 1** (Exponential Convergence Rate of PBA). *For the PBA estimator $\widehat{\theta}_n$, we have*

$$\mathbb{E}|\widehat{\theta}_n - \theta^*| \leq 3e^{-Cn},$$

*where $C > 0$ is a constant of $p$ only.*

**Key Challenges and Contributions.** We assume that $\theta^* \in (0,1)$ for illustration purpose. The basic idea is to show that $\widehat{\theta}_n$ lies within a small interval around $\theta^*$ with high probability. Partition $[0,1]$ into $K$ intervals $[(i-1)/K, i/K), i = 1, 2, \ldots, K$. Then there exists some $i^*$ such that $\theta^* \in [\delta_{i^*-1}, \delta_{i^*})$. We can prove that

$$\mathbb{P}\big(\widehat{\theta}_n \in [\delta_{i^*-1}, \delta_{i^*})\big) \geq 1 - 2K^2(K+1)e^{-Cn}. \quad (1)$$

Choosing $K = e^{Cn/4}$ yields the desired result.

Eq. 1 is equivalent to showing that the PBA estimator is unlikely to be much larger or smaller than $\theta^*$. By symmetry, it suffices to prove the upper-tail bound:

$$\mathbb{P}\big(\widehat{\theta}_n > \delta_{i^*}\big) \leq K^2(K+1)e^{-Cn}. \quad (2)$$

The key challenge is to establish the exponential decay result in Eq. 2.

We emphasize that although similar bounds were obtained in (Burnashev & Zigangirov, 1974; Waeber et al., 2013), their proofs rely on the argument that

$$\mathbb{E}(M_{i+1} - M_i \mid M_i) \leq -C, \quad (3)$$

where $M_i$ is a quantity, in particular, $\log(M_\theta(i))$ in Hero et al. (2007, Theorem 8.1) and $\log(A_i \wedge (1 - A_i))$ in Waeber et al. (2013, Proposition 5.3), that can upper bound $\mathbb{P}\big(\widehat{\theta}_i > \delta_{i^*}\big)$. That is, under the discretization or the Bayesian setting, there exists a stochastic process $M_i$'s that is equivalent to a geometric random walk with negative shift. Hence, Eq. 3 guarantees that the estimator moves closer to the truth after each query (in expectation), regardless of query location. However, our analysis reveals that this property fails for

the original PBA: **the accuracy improvement depends critically on the query position, and improvement is not always guaranteed.**

To overcome this challenge, we conduct a finer-grid analysis of PBA's query dynamics, as detailed in the next subsection. Consequently, our proof of Eq. 2 employs a fundamentally different argument from those in (Burnashev & Zigangirov, 1974; Waeber et al., 2013), which constitutes a key methodological contribution of this work.

## 2.3. Analysis of Query Behaviors

This subsection introduces two novel propositions on query behaviors, which are key for deriving Eq. 2. First, we recall a key property of PBA query: it is the median of posterior belief, meaning that $X_{n+1}$ satisfies

$$\mathbb{P}_n(X_{n+1}) = 1/2. \quad (4)$$

This equation establishes a direct connection between the query location and the posterior probability mass over intervals, which will play a critical role in our analysis.

We introduce the following critical quantities before presenting the results. Let $\delta$ be a constant such that $\theta^* < \delta < 1$. We divide the interval $[0,1]$ into three sub-intervals:

$$I_1 := [0, \theta^*], I_2 := (\theta^*, \delta), I_3 := [\delta, 1],$$

and define

$$a_i^{(j)}(\delta) := \mathbb{P}_i(X \in I_j) := \int_{I_j} f_i(x)dx.$$

We will omit the dependence on $\delta$ when clear from context. Namely, $a_i^{(j)}$ is the posterior probability that the estimator $\widehat{\theta}_{i+1}$ lies in the $j$-th sub-interval after the $i$-th query.

*Remark* 2 (Motivation for $a_i^{(j)}$). At each round $i$, the query must fall into one of three sub-intervals, which is completely determined by $a_{i-1}^{(j)}$'s. Recall that $\widehat{\theta}_n = X_{n+1}$. By Eq. 4, a large estimator $X_{n+1} > \delta$ implies $a_n^{(3)} = \int_\delta^1 f_n(x)dx > 1/2$. Thus, to establish that the probability of such a large estimator is exponentially small, it suffices to show that $a_n^{(3)}$ is unlikely to become large. It turns out that understanding the behavior of PBA queries is necessary for deriving such a result, which further relies on tracking the change of all three posterior probabilities.

For any realization of $X_i, Y_i$'s, we further define

$$N_j := \sum_{i=1}^{n} \mathbb{1}_{X_i \in I_j}, G_j := \{i \in [1,n] : X_i \in I_j\}, j = 1, 2, 3.$$

$N_j$ is the number of total occurrences of the event $X_i \in I_j$, and $G_j$ contains the corresponding indices. We also define

the following stopping times: $\tau_0 = 0$, and for $i = 1, 2, \ldots,$

$$\tau_i = \inf\left\{t : t > \tau_{i-1}, \text{sign}(a_t^{(1)} - \frac{1}{2}) \neq \text{sign}(a_{t-1}^{(1)} - \frac{1}{2})\right\},$$

where $\text{sign}(x) = 1$ for $x \geq 0$ otherwise $\text{sign}(x) = -1$. Namely, $\tau_i$ is the $i$-th time such that $a_t^{(1)}$ across $1/2$, meaning that the query's location transits from $I_1$ to $I_2 \cup I_3$ or vice versa. Finally, we define

$$T := \sup\{i : i \geq 0, \tau_i \leq n\}, \tag{5}$$

which is the number of total times that the query crosses the truth $\theta^*$.

**Proposition 1.** *Let $M_i(\delta) := a_i^{(3)}(\delta)/a_i^{(2)}(\delta)$. For some constant $C_1, C_2 > 0$ of $p$ only, we have*

$$\mathbb{P}(M_n/M_0 \leq e^{-C_1 n}) \geq 1 - e^{-C_2 n},$$

*which implies $\mathbb{E}M_n \leq e^{-C_3 n}/|\theta^* - \delta|$ for some positive constant $C_3$.*

**Proposition 2.** *There exists a constant $\eta$ only depending on $p$, such that $\mathbb{E}(T) \geq \eta n$.*

**Implications.** Together, Propositions 1 and 2 yield a sharp picture of the dynamics: the posterior distribution of PBA rapidly concentrates around $\theta^*$, while the queries themselves are expected to oscillate across the truth. In other words, the queries repeatedly swing around $\theta^*$ but with steadily shrinking amplitude, driving convergence. This insight provides a fundamental explanation for the empirical success of PBA.

Regarding the convergence rate, Proposition 1 shows that $M_n$ decays exponentially fast, which immediately implies an exponentially decaying probability of a large estimator. To see it, Markov's inequality gives that

$$\mathbb{P}(a_n^{(3)} \geq \epsilon) \leq \mathbb{P}(M_n \geq \epsilon) \leq \frac{\mathbb{E}M_n}{\epsilon} \leq \frac{e^{-Cn}}{|\theta^* - \delta|\epsilon}.$$

As a result, $\mathbb{P}(X_{n+1} \geq \delta) \leq \mathbb{P}(a_n^{(3)} \geq 1/2) \leq 2e^{-Cn}/|\theta^* - \delta|$, establishing the kep step (2) in Theorem 1.

**Proof Sketch.** The core idea behind the proof of Proposition 1 is to show that $\ln(M_i)$'s form a supermartingale, which decreases when the query lies in $I_2$ or $I_3$. We note that $\ln(M_i)$ remains unchanged when $X_i \in I_1$, and the decrease can be arbitrarily small when $X_i \in I_2$. Fortunately, we find that $\ln(M_i)$ decreases by at least a constant amount when the query crosses the truth. That is, when $X_{i-1} < \theta^* \leq X_i$ or $X_{i-1} \geq \theta^* > X_i$. This boundary-crossing behavior is characterized by $T$. Hence, to ensure that $\ln(M_n)$ becomes sufficiently small, it suffices to show that $\mathbb{E}(T)$ grows linearly with $n$, as established in Proposition 2.

Technically, Propositions 1 and 2 hinge on a careful analysis of the changes in $a_i^{(j)}$ and their combinations such as

$M_i$. These changes depend on the query location and leads to a discussion of three cases: $X_i \in I_1$, $X_i \in I_2$, and $X_i \in I_3$. To prove Proposition 2, first we need to construct appropriate sub- or super-martingales from the posterior probabilities. We then show that the queries cross the truth sufficiently often by analyzing the boundary-crossing times $\tau_i$ and invoking the stopping time theorem. This, together with a martingale concentration inequality, ensures a significant reduction in $M_i$, thereby completing the proof of Proposition 1. The full details of these two propositions are presented below.

*Remark* 3 (Motivation of $M_i$). Proposition 1 focuses on analyzing $M_n$ rather than $a_n^{(3)}$. The quantity $M_n$ is deliberately and carefully designed, not an arbitrary combination of the $a_n^{(j)}$'s. The key reason is that the evolution of $a_n^{(j)}$ depends intricately on the query locations, making them difficult to control directly. To establish Eq. 2, we seek a quantity that is guaranteed to be monotone on average across rounds. However, the $a_n^{(j)}$'s alone do not exhibit this property for all possible query positions. By introducing a ratio-based structure, $M_n$ (specifically, its logarithm) acquires this desirable monotonicity, enabling a tractable analysis.

**Proof of Proposition 1.**

*Proof.* We prove this result by three steps: (1) $M_i$ is expected to decrease or maintain the same at each round, (2) there is a sufficient number of time steps such that $M_i$ is expected to decrease, (3) evoking a concentration inequality.

**Step 1:** Depending on the position of $X_i$, we discuss the update of $M_i$ in three cases as follows.

**Case 1, $X_i \in I_1$.** Clearly, by the update rule of PBA, $a^{(2)}$ and $a^{(3)}$ will be multiplied by $2(1-p)$ (when $Y_i = 0$) or $2p$ (when $Y_i = 1$) simultaneously. As a result, $M_i = M_{i-1}$ in this case.

**Case 2, $X_i \in I_2$.** Now, a correct label ($Y_i = 1$ with probability $1-p$) leads to $a_i^{(1)} = 2(1-p)a_{i-1}^{(1)}$ and $a_i^{(3)} = 2pa_{i-1}^{(3)}$. As a result,

$$M_i/M_{i-1} = \frac{2pa_{i-1}^{(2)}}{1 - 2(1-p)a_{i-1}^{(1)} - 2pa_{i-1}^{(3)}} \in \left(\frac{p}{1-p}, 1\right).$$

A wrong label leads to

$$M_i/M_{i-1} = \frac{2(1-p)a_{i-1}^{(2)}}{1 - 2pa_{i-1}^{(1)} - 2(1-p)a_{i-1}^{(3)}} \in \left(1, \frac{1-p}{p}\right).$$

For notation simplicity, we denote $q_1 = a_{i-1}^{(1)}, q_2 = a_{i-1}^{(2)}, q_3 = a_{i-1}^{(3)}$. Some important properties of them are

summarized in Lemma 1. Evoking Lemma 1, we have

$$\mathbb{E}\left(\frac{M_i}{M_{i-1}}\right) = (1-p)\frac{2pq_2}{1-2(1-p)q_1-2pq_3}$$
$$+ p\frac{2(1-p)q_2}{1-2pq_1-2(1-p)q_3}$$
$$= \frac{2p(1-p)q_2}{q_2-(1-2p)(q_1-q_3)} + \frac{2p(1-p)q_2}{q_2+(1-2p)(q_1-q_3)}$$
$$= \frac{4p(1-p)(q_2)^2}{(q_2)^2-(1-2p)^2(q_1-q_3)^2}$$
$$= 1 - \frac{(1-2p)^2\{(q_2)^2-(q_1-q_3)^2\}}{(q_2)^2-(1-2p)^2(q_1-q_3)^2}$$
$$< 1.$$

The last step is due to $q_2 > |q_1 - q_3|$ and the positivity of denominator.

Moreover, for any $\epsilon \in (0, 1/2)$, when $q_1, q_3 \le (1-\epsilon)/2$, the fourth point of Lemma 1 gives that $q_2 - |q_1 - q_3| \ge \epsilon$ and

$$\mathbb{E}\left(\frac{M_i}{M_{i-1}}\right) = 1 - \frac{(1-2p)^2\{(q_2)^2-(q_1-q_3)^2\}}{(q_2)^2-(1-2p)^2(q_1-q_3)^2}$$
$$\le 1 - (1-2p)^2\epsilon^2.$$

As a result, Jensen's Inequality gives

$$\mathbb{E}\left\{\ln\left(\frac{M_i}{M_{i-1}}\right)\right\} \le \ln(1-(1-2p)^2\epsilon^2) \le -(1-2p)^2\epsilon^2.$$

**Case 3, $X_i \in I_3$.** In this case, a correct label ($Y_i = 1$ with probability $1-p$) leads to $a_i^{(2)} = 2(1-p)a_{i-1}^{(2)}$ and $1 - a_i^{(3)} = 2(1-p)(1-a_{i-1}^{(3)})$, so that

$$M_i/M_{i-1} = \frac{1-2(1-p)(1-a_{i-1}^{(3)})}{2(1-p)a_{i-1}^{(3)}} \in \left(\frac{p}{1-p}, \frac{1}{2(1-p)}\right).$$

Similarly, a wrong label results in

$$M_i/M_{i-1} = \frac{1-2p(1-a_{i-1}^{(3)})}{2pa_{i-1}^{(3)}} \in \left(\frac{1}{2p}, \frac{1-p}{p}\right).$$

As a result, we have

$$\mathbb{E}\left\{\ln\left(\frac{M_i}{M_{i-1}}\right)\right\} = (1-p)\ln\left(\frac{1-2(1-p)(1-a_{i-1}^{(3)})}{2(1-p)a_{i-1}^{(3)}}\right)$$
$$+ p\ln\left(\frac{1-2p(1-a_{i-1}^{(3)})}{2pa_{i-1}^{(3)}}\right).$$

Let

$$h(x) := (1-p)\ln\left(\frac{1-2(1-p)(1-x)}{x}\right)$$
$$+ p\ln\left(\frac{1-2p(1-x)}{x}\right).$$

Its first derivative is

$$h'(x) = \frac{2(1-p)^2}{1-2(1-p)(1-x)} + \frac{2p^2}{1-2p(1-x)} - \frac{1}{x}.$$

We have

$$h'(x) > 0 \iff 2(1-p)^2x + 2p^2x$$
$$> \{1-2(1-p)(1-x)\}\{1-2p(1-x)\}$$
$$\iff (2-4p+4p^2)x > 1 - 2(1-x) + 4p(1-p)(1-x)$$
$$\iff 2x > 2x - 1 + 4p(1-p)$$
$$\iff 1 > 4p(1-p),$$

which is true since $p \in (0, 1/2)$. Since $\mathbb{E}\left\{\ln\left(\frac{M_i}{M_{i-1}}\right)\right\} = h(a_{i-1}^{(3)}) - \ln(2) + H(p)$ where $H(p) = -p\ln(p) - (1-p)\ln(1-p)$ is the binary entropy function, we know that $\mathbb{E}\left\{\ln\left(\frac{M_i}{M_{i-1}}\right)\right\}$ achieves its maximum when $a_{i-1}^{(3)} = 1$, leading to

$$\mathbb{E}\left\{\ln\left(\frac{M_i}{M_{i-1}}\right)\right\} \le -\ln(2) + H(p) < 0.$$

**Step 2:** We show that $M_i$ is expected to strictly decrease for sufficient number of rounds. Specifically, we know that $\mathbb{E}\left\{\ln\left(\frac{M_i}{M_{i-1}}\right)\right\}$ is strictly smaller than zero if (1) $X_i \in I_2$, and $q_1, q_3 < (1-\epsilon)/2$; and (2) $X_i \in I_3$. For a given realization of $X_i, Y_i$'s, the latter case happens for $N_3$ times. We define the number of the first case as $N_2'(\epsilon) := |G_2'(\epsilon)|$, where

$$G_2'(\epsilon) := \{i : X_i \in I_2, q_1, q_3 \le (1-\epsilon)/2\}.$$

We will omit $\epsilon$ in the following as it will be chosen as a constant of $p$ solely.

Next, we show that $N_2' + N_3 \ge \eta'n$ with high prob for some $\eta'$. The idea is to show that each down-crossing of $\tau_i$ leads to an instance of $G_2'$ or $G_3$ with a constant probability, and Proposition 2 shows that such down-crossing happens sufficiently often. Let us consider each time $a_t^{(1)}$ goes down and crosses $1/2$. Suppose $a_{t-1}^{(1)} > 1/2$ and $a_t^{(1)} \le 1/2$. By update rule, we have $2p \le a_t^{(1)} \le 1/2$, thus either $X_{t+1} \in I_2$ or $X_{t+1} \in I_3$.

(Step 2.1) We have $t + 1 \in G_2' \cup G_3$ when $X_{t+1} \in I_3$ or $X_{t+1} \in I_2$ with $a_t^{(1)}, a_t^{(3)} \le (1-\epsilon)/2$.

(Step 2.2) Now, suppose $t \notin G_2' \cup G_3$, namely $X_{t+1} \in I_2$ and at least one of $a_t^{(1)}, a_t^{(3)}$ is larger than $(1-\epsilon)/2$.

We first consider the case where $a_t^{(1)} > (1-\epsilon)/2$. With probability $p$, $Y_{t+1}$ is a wrong label, and $X_{t+2} \in I_1$ since

$a_{t+1}^{(1)} = 2(1-p)a_t^{(1)} > 1/2$ for any $\epsilon \in (0, 1-1/(2-2p))$. With probability $1-p$, $Y_{t+1}$ is a correct label, leading to (i) $X_{t+2} \in I_3$, or (ii) $X_{t+2} \in I_2$. While (i) automatically leads to $t+2 \in G_3$, (ii) again leads to two possible outcomes: (ii.a) $X_{t+3} \in I_3$, or (ii.b) $X_{t+3} \in I_2$. We note that (ii.b) results in $t+3 \in G_2'$ for a sufficiently small $\epsilon$. To see it, we have $a_{t+2}^{(1)} = 4p(1-p)a_t^{(1)} < (1-\epsilon)/2$ and $a_{t+2}^{(3)} = 4p(1-p)a_t^{(3)} < (1-\epsilon)/2$ for any $\epsilon \in (0, (1-2p)^2)$. Next, we consider the case $a_t^{(3)} > (1-\epsilon)/2$. With probability $p$, $Y_{t+1}$ is a wrong label, we therefore have $t+2 \in N_3$ because $a_{t+1}^{(3)} = 1-2p(1-a_t^{(3)}) > 1-p(1+\epsilon) > 1/2$ for any $\epsilon \in (0, (2p)^{-1}-1)$. Combining these two cases, we have that with probability at least $p$, such time step $t$ will lead to an occurrence of $N_2'$ or $N_3$ before the next occurrence of $a_t^{(1)}$ going down and crossing $1/2$.

(Step 2.3) WLOG, let $a_0^{(1)} < 1/2$ as explained in the proof of Proposition 2. Let $R_k$ denoting whether $a_{\tau_{2k}}^{(1)}$ leads to an occurrence of $N_2'$ or $N_3$, we have $R_k$ being Bernoulli random variables with $\mathbb{P}(R_k = 1) \geq p$. We can therefore construct a sub-martingale $S_l = \sum_{k=1}^{l} R_k - pl$, $l = 1, 2, \ldots$, and $S_0 = 0$. Now, applying the optional stopping theorem, we have $\mathbb{E}S_{\lfloor T/2 \rfloor} \geq \mathbb{E}S_0 = 0$, yielding

$$\mathbb{E}(N_2' + N_3) \geq \mathbb{E}\left(\sum_{k=1}^{\lfloor T/2 \rfloor} R_k\right) \geq p\mathbb{E}(\lfloor T/2 \rfloor).$$

Now, evoking Proposition 2, we have $\mathbb{E}(N_2' + N_3) \geq \eta' n$ for $\eta' = p\eta/2$.

**Step 3:** Finally, we show that $M_n$ is small with high probability by applying Azuma-Hoeffding inequality. Note that

$$M_n = M_0 \exp\left\{\sum_{i=1}^{n} \ln(M_i/M_{i-1})\right\}.$$

Step 1 indicates that $\sum_{i=1}^{n} \ln(M_i/M_{i-1})$ is a supermartingale with respect to $n$, because

$$\mathbb{E}\left\{\sum_{i=1}^{n} \ln(M_i/M_{i-1}) \mid \sum_{i=1}^{n-1} \ln(M_i/M_{i-1})\right\}$$
$$= \mathbb{E}\ln(M_n/M_{n-1}) \leq 0.$$

Moreover, all $\ln(M_i/M_{i-1})$'s have a uniform upper bound on their absolute value and variance, denoted as $B_1, B_2 > 0$, respectively. Let $C_6 := \min\{(1-2p)^2\epsilon^2, \ln(2) - H(p)\} > 0$ and $\zeta = \eta' C_6/2$. By Azuma-Hoeffding's inequality, we have

$$\mathbb{P}\left(\sum_{i=1}^{n} \ln(M_i/M_{i-1}) > \mathbb{E}\left(\sum_{i=1}^{n} \ln(M_i/M_{i-1})\right) + n\zeta\right)$$

is $\leq e^{-2n\zeta^2}$. This inequality holds if and only if

$$\mathbb{P}\left(\sum_{i=1}^{n} \ln(M_i/M_{i-1}) > -\mathbb{E}(N_2' + N_3)C_6 + n\zeta\right) \leq e^{-2n\zeta^2}.$$

Moreover, this condition holds if and only if

$$\mathbb{P}\left(\sum_{i=1}^{n} \ln(M_i/M_{i-1}) > -\eta' C_6 n/2\right) \leq e^{-2n\zeta^2}.$$

As a result, with probability at least $1 - e^{-2n\zeta^2}$, we have

$$M_n \leq M_0 \exp(-n\eta' C_6/2).$$

We therefore complete the proof by noting $M_0 = (1-\delta)/(\delta - \theta^*)$ since $f_0(x) = 1$ for all $x \in [0,1]$. $\qquad\square$

**Proof of Proposition 2.**

*Proof.* Let $b_i^{(1)} = 1 - a_i^{(1)}$. Depending on the position of $X_i$, The change of $a_i^{(1)}$ in each round is also categorized into three cases.

**Case 1,** $X_i \leq \theta^*$. A correct label ($Y = 0$ with probability $1-p$) leads to $1 - a_i^{(1)} = 2(1-p)(1 - a_{i-1}^{(1)})$. Therefore,

$$\frac{a_i^{(1)}}{a_{i-1}^{(1)}} = \frac{1 - 2(1-p)(1 - a_{i-1}^{(1)})}{a_{i-1}^{(1)}}, \quad \frac{b_i^{(1)}}{b_{i-1}^{(1)}} = 2(1-p),$$

A wrong label leads to

$$\frac{a_i^{(1)}}{a_{i-1}^{(1)}} = \frac{1 - 2p(1 - a_{i-1}^{(1)})}{a_{i-1}^{(1)}}, \quad \frac{b_i^{(1)}}{b_{i-1}^{(1)}} = 2p.$$

Therefore,

$$\mathbb{E}\left\{\ln\left(\frac{a_i^{(1)}}{a_{i-1}^{(1)}}\right)\right\} = (1-p)\ln\left(\frac{1 - 2(1-p)(1 - a_{i-1}^{(1)})}{a_{i-1}^{(1)}}\right)$$
$$+ p\ln\left(\frac{1 - 2p(1 - a_{i-1}^{(1)})}{a_{i-1}^{(1)}}\right) < 0.$$

The last inequality is because function $h(a_{i-1}^{(1)}) := (1-p)\ln\left(\frac{1-2(1-p)(1-a_{i-1}^{(1)})}{a_{i-1}^{(1)}}\right) + p\ln\left(\frac{1-2p(1-a_{i-1}^{(1)})}{a_{i-1}^{(1)}}\right)$ monotonously increases when $a_{i-1}^{(1)} \in (1/2, 1)$, which can be verified by taking its first derivative.

Also,

$$\mathbb{E}\left\{\ln\left(\frac{b_i^{(1)}}{b_{i-1}^{(1)}}\right)\right\} = (1-p)\ln(2(1-p)) + p\ln(2p)$$
$$= \ln(2) - H(p) > 0,$$

where $H(p) = -p \ln(p) - (1-p) \ln(1-p)$ is the binary entropy function.

**Case 2, $X_i \geq \delta$.** A correct label ($Y = 1$ with probability $1 - p$) leads to $a_i^{(1)} = 2(1 - p)a_{i-1}^{(1)}$ and

$$\frac{a_i^{(1)}}{a_{i-1}^{(1)}} = 2(1 - p), \quad \frac{b_i^{(1)}}{b_{i-1}^{(1)}} = \frac{1 - 2(1 - p)(1 - b_{i-1}^{(1)})}{b_{i-1}^{(1)}}.$$

A wrong label leads to

$$\frac{a_i^{(1)}}{a_{i-1}^{(1)}} = 2p, \quad \frac{b_i^{(1)}}{b_{i-1}^{(1)}} = \frac{1 - 2p(1 - b_{i-1}^{(1)})}{b_{i-1}^{(1)}},$$

Therefore,

$$\mathbb{E}\left\{\ln\left(\frac{a_i^{(1)}}{a_{i-1}^{(1)}}\right)\right\} = (1 - p)\ln\{2(1 - p)\} + p \ln(2p)$$

$$= \ln(2) - H(p) > 0.$$

Thus, $\mathbb{E}\left\{\ln\left(\frac{b_i^{(1)}}{b_{i-1}^{(1)}}\right)\right\} < 0.$

**Case 3, $\theta^* < X_i < \delta$.** The update rule for $a_i^{(1)}$ is exactly the same as Case 2, hence we have

$$\mathbb{E}\left\{\ln\left(\frac{a_i^{(1)}}{a_{i-1}^{(1)}}\right)\right\} = (1 - p)\ln\{2(1 - p)\} + p \ln(2p)$$

$$= \ln(2) - H(p).$$

Hence, $\mathbb{E}\left\{\ln\left(\frac{b_i^{(1)}}{b_{i-1}^{(1)}}\right)\right\} < 0$. For now, we assume that $2p < a_0^{(1)} < 1/2$; otherwise we can start the count of $N_1$ at the first time $t$ such that $2p < a_t^{(1)} < 1/2$, as explained later. Therefore, $\tau_{2k-1}, k = 1, 2, \ldots$ is the time that $a_t^{(1)}$ goes up and crosses $1/2$, while $\tau_{2k}, k = 1, 2, \ldots$ is the time that $a_t^{(1)}$ goes down and crosses $1/2$, and $T$ is the number of total cross times.

We note that $Z_i := \tau_i - \tau_{i-1}, i = 1, 2, \ldots$ are random variables with $\mathbb{E}Z_i \leq z$, where $z$ is a constant. To see it, we have

$$a_{\tau_{2k-1}}^{(1)} = a_{\tau_{2k-2}}^{(1)} \exp\left\{\sum_{i=0}^{\tau_{2k-1} - \tau_{2k-2}} \ln\left(\frac{a_{\tau_{2k-2}+i}^{(1)}}{a_{\tau_{2k-2}+i-1}^{(1)}}\right)\right\}$$

$$\geq \exp\left(\sum_{i=1}^{\tau_{2k-1} - \tau_{2k-2}} V_i\right)/(2p),$$

$$b_{\tau_{2k}}^{(1)} = b_{\tau_{2k-1}}^{(1)} \exp\left\{\sum_{i=0}^{\tau_{2k} - \tau_{2k-1}} \ln\left(\frac{b_{\tau_{2k-1}+i}^{(1)}}{b_{\tau_{2k-1}+i-1}^{(1)}}\right)\right\}$$

$$\leq 2(1 - p)\exp\left(-\sum_{i=1}^{\tau_{2k} - \tau_{2k-1}} V_i\right), \quad (6)$$

where $V_i$'s are independent random variable with $\mathbb{E}V_i = \ln(2) - H(p) := v$. Moreover, $V_i$'s are uniformly bounded by a constant of $p$ solely, denoted by $B$. Therefore, $\ln(a_t^{(1)})$ is a random walk starting from (or above) $-\ln(2p)$ with a positive drift, which is expected to across $\ln(1/2)$ in a finite time by random walk theory (can be easily verified by applying Hoeffding's inequality). Similarly, $\ln(b_t^{(1)})$ is a random walk starting from (or below) $\ln(2(1 - p))$ with a negative drift. We further define $S_l = \sum_{k=1}^{l} Z_k - kz$ and $S_0 = 0$. Clearly, $S_l$ is a super-martingale. Finally, optional stopping theorem yields that $\mathbb{E}S_T \leq S_0$, leading to

$$\mathbb{E}(T + 1)z \geq \mathbb{E}\left\{\sum_{k=1}^{T+1}(\tau_k - \tau_{k-1})\right\} = \mathbb{E}(\tau_{T+1}) \geq n.$$

As a result, we have $\mathbb{E}(T) \geq \eta n$ for $\eta = 1/(2z)$.

Finally, we show that we can assume $2p < a_t^{(1)} < 1/2$. By Hoeffding's inequality, $a_t^{(1)}$ will across $1/2$ both up and down at least once within $n/2$ steps with probability at least $1 - e^{-C_4 n}$ with some constant $C_4$. Ever since that, we will have $2p \leq a_{\tau_i}^{(1)} < 1/2$ when $\text{sign}(a_{\tau_i}^{(1)}) = -1$ and $1/2 \leq a_{\tau_i}^{(1)} < 2(1 - p)$ when $\text{sign}(a_{\tau_i}^{(1)}) = 1$, due to the update rule. We therefore conclude the proof. $\square$

## 3. Extension to High Dimensional Data

In this section, we extend our results to high dimensional setting where $d \geq 2$.

**Setup.** Consider the query $X \in [0, 1]^d, d \geq 2$ and the label $Y \in \{0, 1\}$. Similar to the setting when $d = 1$, let $p \in (0, 1/2)$ represent the noise level in the labels, $\mathbb{P}(Y = h(X)) = 1 - p$ and $\mathbb{P}(Y = 1 - h(X)) = p$, where $h$ is a classifier $h : [0, 1]^d \rightarrow \{0, 1\}$, which a learner wants to estimate. Recall that in one dimensional setting, we consider a hypothesis class $\mathcal{H} = \{h_\theta : \theta \in [0, 1]\}$ and work with a threshold classifier $h_{\theta^*}(x) = \mathbb{1}_{x \geq \theta^*}$. This ordered, one-parameter structure enables a probabilistic bisection algorithm (PBA, see Section 2), yielding an estimator $\widehat{\theta}_n$ which converges to $\theta^*$ exponentially fast, i.e. $\mathbb{E}|\widehat{\theta}_n - \theta^*| \leq \mathcal{O}(e^{-n})$ (see Theorem 1).

For $d \geq 2$, the natural analogue of a "threshold" is a decision boundary, whose shape should be restricted by additional geometric assumptions, such as smoothness, to ensure identifiability and control the complexity of the hypothesis class. In this work, we adopt a standard assumption in the literature that the decision boundary is Hölder smooth (Castro & Nowak, 2007; 2008). In particular, we consider the hypothesis class $\mathcal{H} = \{h_g : g \in \Sigma(L, \alpha)\}$, where $\Sigma(L, \alpha)$ denotes $\alpha$-Hölder smooth with parameters $L$ (see Definition 1).

**Definition 1.** A function $g : [0, 1]^{d-1} \rightarrow \mathbb{R}$ is Hölder smooth if it has continuous partial derivatives up to order

$k = \lfloor \alpha \rfloor$ and $\forall \boldsymbol{z}, \boldsymbol{x} \in [0,1]^{d-1}$, $g(\boldsymbol{z}) - \mathrm{TP}_{\boldsymbol{x}}(\boldsymbol{z}) \leq L\|\boldsymbol{z} - \boldsymbol{x}\|^{\alpha}$, where $L, \alpha > 0$, and $\mathrm{TP}_{\boldsymbol{x}}(\cdot)$ denotes the order $k$ Taylor polynomial approximation of $g$ expanded around $\boldsymbol{x}$.

The classifier is $h_{g^*}(x) = \mathbb{1}_{x \in G^*}$, where $g^*$ is the decision boundary of $G^*$ and $G^* = \{(\widetilde{X}, x_d) \in [0,1]^{d-1} \times [0,1] : x_d \geq g^*(\widetilde{X})\}$. In the following, we use $h$, $g^*$, and $G^*$ interchangeably. The learner wants to construct an estimator $\widehat{g}_n$, or equivalently, a classifier $\widehat{G}_n = \{(\widetilde{X}, x_d) \in [0,1]^{d-1} \times [0,1] : x_d \geq \widehat{g}_n(\widetilde{X})\}$, with small expected $L_1$ error $\mathbb{E}\|\widehat{g}_n - g^*\|_1$.

**Theorem 2.** *There exists an estimator $\widehat{g}_n$ such that $\mathbb{E}\|\widehat{g}_n - g^*\|_1 \leq \mathcal{O}\left( \left( \frac{\log n}{n} \right)^{\frac{\alpha}{d-1}} \right)$.*

We highlight that in the high-dimensional setting, the upper bound in Theorem 2 is **nearly minimax optimal**, since the matching information-theoretic lower bound of Castro & Nowak (2008) for learning Hölder-smooth decision boundaries implies that no estimator can achieve $L_1$ error smaller than a constant multiple of $n^{-\frac{\alpha}{(d-1)}}$ (up to logarithmic factors).

In the proof (see Appendix Subsection A.2), we explicitly construct $\widehat{g}_n$ by generalizing the PBA to $d \geq 2$. At a high level, we recursively partition the $(d-1)$-dimensional base domain into dyadic cells and on each vertical lines, we deploy a one-dimensional PBA to localize the decision boundary within each cell. By combining these local estimates across the cells, we obtain a piecewise approximation of the boundary. The Hölder regularity of $g^*$ governs both the approximation error within each cell and the number of cells required at a given resolution, allowing a sample allocation that achieves the convergence rate in Theorem 2.

**Special case of linear decision boundary.** A linear decision boundary is arbitrarily smooth, corresponding to the special case of $\alpha = \infty$. Theorem 2 implies that learning such a function using a PBA-based algorithm is faster than any polynomial rate. In fact, Theorem 3 demonstrates that an **optimal exponential convergence rate** can still be achieved by leveraging the PBA. To the best of our knowledge, **this is the first result establishing such a guarantee.**

**Theorem 3.** *When the true boundary $g^*$ is linear, corresponding to the case $\alpha = \infty$, for any sufficiently large $n$, there exists an estimator $\widehat{g}_n$ satisfying $\mathbb{E}\|\widehat{g}_n - g^*\|_1 \leq C_1 \exp(-cn)$, where $C_1 > 0$ depending only on $d$ and $c > 0$ depending only on $d$ and the noise level $p$.*

When the decision boundary is linear, the proof proceeds in two stages. In $d$ dimensions, a linear decision boundary induces a hyperplane that intersects a subset of the $d2^{d-1}$ edges of the unit cube $[0,1]^d$. Each such intersection is characterized by a change in the label distribution along the corresponding edge.

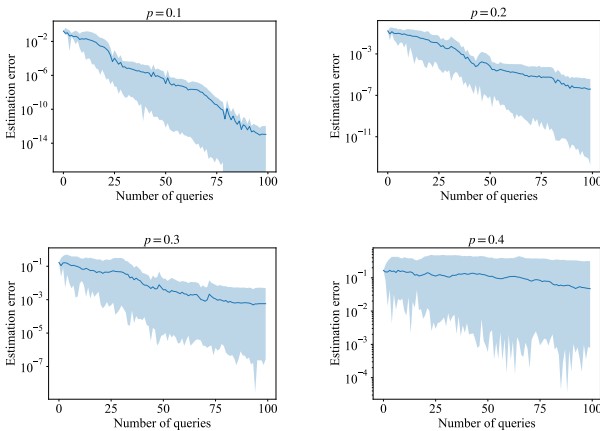

*Figure 1.* Estimator error rate of PBA estimator with respect to the query size $n$, under various noise level $p$.

In the first step, we perform edge detection via hypothesis testing. For each edge of the cube, we query the two endpoints and test whether the conditional label distributions differ across the edge. Using a two-sample test and a union bound over all $d2^{d-1}$ edges, we show that, with probability at least $1 - C \exp(-cn)$, with some positive constant $C$ and $c$, the procedure correctly identifies the entire set of edges intersected by the decision boundary, while making no false discoveries.

In the second step, for each detected edge, we localize the intersection point by running PBA along the edge to estimate the threshold at which the label switches. Finally, we reconstruct the hyperplane parameters by solving a linear regression problem using these estimated intersection points. Combining the exponential localization error of PBA with the stability of the linear reconstruction yields an overall exponential convergence rate for the estimated decision boundary. We direct readers to Subsection A.3 for details. The pseudo-code of the PBA estimator $\widehat{g}_n$ in the high-dimensional case is provided in Appendix C.

## 4. Experiments

In this section, we conducted simulation experiments to corroborate our theoretical findings.

We start with the one-dimensional case. WLOG, we choose $\theta^* = 1/3$ and vary the noisy level $p$ from a list of values $0.1, 0.2, 0.3, 0.4$. We report the average estimation error of the PBA estimator with respect to the query size $n$ on 20 replicated experiments. The results are shown in Figure 1. The maximum query size is 100 because the convergence rate is exponentially fast and the calculation of estimation error will encounter numerical issues, as seen in Figure 1.

**Figure 1 clearly displays an exponential decay of the**

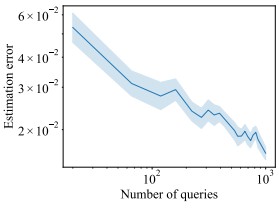

*Figure 2.* Estimator error rate of PBA estimator for a Lipschitz boundary, with respect to the query size $n$. Both x- and y-axis are plotted on a logarithmic scale.

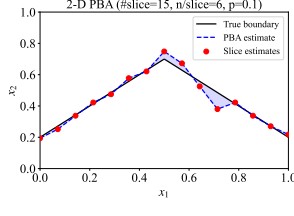

*Figure 3.* Comparison of true Lipschitz boundary and estimated boundary by PBA.

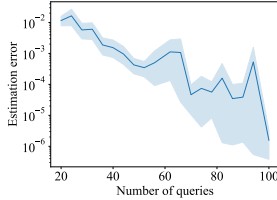

*Figure 4.* Estimator error rate of PBA for a linear boundary with respect to the query size $n$. The y-axis is plotted on a logarithmic scale.

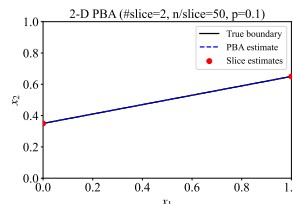

*Figure 5.* Comparison of true linear boundary and estimated boundary by PBA.

estimation error by PBA (a linear trend in the log-plot), aligned with our Theorem 1. In addition, a larger noise level $p$ yields a substantially smaller constant in the exponent of the convergence rate, resulting in slower convergence. This confirms that the constant $C$ in Theorem 1 depends on $p$.

We then conduct experiments to empirically validate Theorems 2 and 3. First, we apply PBA to learn a Lipschitz continuous boundary. We report (1) the average estimation error as a function of the query budget $n$ over 50 independent runs in Figure 2, and (2) a visualization comparing the true and estimated boundaries in Figure 3. The noise level is set to $p = 0.1$, and the number of vertical slices is chosen as $M = \lfloor n/\log(n) \rfloor$, following our theoretical guidance.

Next, we consider learning a linear boundary $g(x) = 0.35 + 0.3x_1$, where $x_1$ denotes the first coordinate of $x$. In this setting, PBA only needs to query two slices, leading to an exponential convergence rate. The results are shown in Figures 4 and 5.

Overall, the experimental results are consistent with our theoretical findings: exponential convergence is generally not achievable for high-dimensional functions, except in special cases such as linear classifiers.

## 5. Conclusion and Further Remarks

This work investigates the dynamics of PBA queries, revealing the intriguing pattern that they oscillate around the truth while steadily converging toward it. Building on this insight, we establish the exponential convergence rate of PBA, thereby bridging the long-standing gap between its theoretical guarantees and empirical performance. A natural direction for future research is to examine whether PBA still converges exponentially when the actual noise level $p$ exceeds the one assumed in the update rule, and, if not, to determine the resulting convergence rate. Another intriguing problem is the implementation of PBA, as it may be numerically challenge to exactly find the posterior median.

## Impact Statement

This paper presents work whose goal is to advance the field of machine learning. There are many potential societal consequences of our work, none of which we feel must be specifically highlighted here.

## Acknowledgment

This work is partially supported by the ONR Award N00014-23-1-2802.

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

## A. Missing Proofs.

### A.1. Proof of Theorem 1.

*Proof.* We handle the case with $\theta^* \in (0, 1)$ first, and defer proof of the corner case to the end of this proof. For now, let $\delta$ be a constant such that $1 > \delta > \theta^* > 0$. We denote

$$M_i(\delta) := \frac{a_i^{(3)}(\delta)}{a_i^{(2)}(\delta)}. \tag{7}$$

Proposition 1 shows that for some constant $C > 0$ of $p$ only, we have

$$\mathbb{E}M_n \leq \frac{e^{-Cn}}{|\theta^* - \delta|}. \tag{8}$$

Since $a_i^{(j)} \in (0, 1)$ for $j = 1, 2, 3$ and all $i \geq 0$, we know that $M_n$ is positive, and Markov's inequality gives that

$$\mathbb{P}(a_n^{(3)} \geq \epsilon) \leq \mathbb{P}(M_n \geq \epsilon) \leq \frac{\mathbb{E}M_n}{\epsilon} \leq \frac{e^{-Cn}}{|\theta^* - \delta|\epsilon}.$$

When $0 < \delta < \theta^* < 1$, we can apply Proposition 1 after performing the transformation $x' = 1 - x$, which yields $\mathbb{P}(1 - a_n^{(3)} \geq \epsilon) \leq e^{-Cn}/(|\theta^* - \delta|\epsilon)$. Therefore,

$$\mathbb{P}(\min\{a_n^{(3)}, 1 - a_n^{(3)}\} \geq \epsilon) \leq e^{-Cn}/(|\theta^* - \delta|\epsilon). \tag{9}$$

When $\theta^* = 0$ or $\theta^* = 1$, we have $\mathbb{P}(\min\{a_n^{(3)}, 1 - a_n^{(3)}\} \geq \epsilon) = 0$ by definition (see, Eq. 7), therefore satisfying Eq. 9 as well.

Now, let $\delta_i = i/K, i = 0, \ldots, K$, where $K$ will be determined shortly. If $\min_i |\theta^* - \delta_i| < 1/\{2K(K+1)\}$, we can increase $K$ by 1, which ensures that $\min_i |\theta^* - \delta_i| \geq 1/\{2K(K+1)\}$. Clearly, there exists some $i^* \geq 1$ such that $\theta^* \in (\delta_{i^*-1}, \delta_{i^*})$. Evoking Eq. 9, we know that with probability at least $1 - 2K(K+1)e^{-Cn}/\epsilon$, we have

$$\mathbb{P}_n(X \in (\delta_{i^*-1}, \delta_{i^*})) \geq 1 - 2\epsilon,$$

implying that $|X_{n+1} - \theta^*| \leq 1/K$ for any $\epsilon < 1/4$. Therefore, for $K > 4$ and $\epsilon = 1/K$, we have

$$\mathbb{P}(|X_{n+1} - \theta^*| > 1/K) \leq 2K^2(K+1)e^{-Cn}.$$

Finally, taking $K = e^{Cn/4}$ yields

$$\mathbb{E}|X_{n+1} - \theta^*| \leq 1/K + \mathbb{P}(|X_{n+1} - \theta^*| > 1/K) \leq 3e^{-Cn/4}.$$

Regarding the corner case, we analyze with $\theta^* = 0$ as $\theta^* = 1$ can be handled with an analogous argument. In this case, Lemma 2 gives that

$$\mathbb{E}(a_n^{(3)}) \leq e^{-Cn}.$$

Let $\delta = \epsilon = 1/K$. Similar to the argument in the case of $\theta^* \in (0, 1)$, we have with probability at least $1 - e^{-Cn}/\epsilon$,

$$\mathbb{P}_n(X \in [0, \delta]) \geq 1 - \epsilon.$$

Choosing $K = e^{Cn/2}$ gives

$$\mathbb{E}|X_{n+1} - \theta^*| \leq 1/K + \mathbb{P}(|X_{n+1} - \theta^*| > 1/K) \leq 2e^{-Cn/2}.$$

We thus conclude the proof. $\qquad\square$

**Lemma 1.** *When $X_i \in (\theta^*, \delta)$, we have the following facts: (1) $q_1 + q_2 + q_3 = 1$. (2) $q_1, q_3 \in (0, 1/2)$. (3) $|q_1 - q_3| < q_2$. (4) For any $\epsilon < 1/2$, $q_2 - |q_1 - q_3| \geq \epsilon$ if and only if $q_1, q_3 \leq (1 - \epsilon)/2$.*

*Proof.* Fact (1) is by definition of $a_{i-1}^{(j)}$, $j = 1, 2, 3$. Their summation equals to $\mathbb{P}_{i-1}(X \leq 1) = 1$.

Fact (2) holds since $X_i \in (\theta^*, \delta)$; otherwise, if $q_1 \geq 1/2$ for example, we have $X_i \leq \theta^*$ since $P_{i-1}(X < \theta^*) = q_1 \geq 1/2$, which is a contradiction.

We prove Fact (3) by contradiction. If $|q_1 - q_3| \geq q_2$, then $q_1 \geq q_2 + q_3$ or $q_3 \geq q_1 + q_2$. However, $q_1 \geq q_2 + q_3$ with Fact (1) imply that $q_1 \geq 1/2$, which is a contradiction to Fact (2). Similarly, $q_3 \geq q_1 + q_2$ cannot hold as well.

Regarding (4), we use a similar argument as (3). Note that

$$q_2 - |q_1 - q_3| < \epsilon$$
$$\Longleftrightarrow q_1 > q_2 + q_3 - \epsilon \text{ or } q_3 > q_2 + q_1 - \epsilon$$
$$\Longleftrightarrow q_1 > (1 - \epsilon)/2 \text{ or } q_3 > (1 - \epsilon)/2.$$

We thus complete the proof. □

**Lemma 2.** *When $\theta^* = 0$ and $\delta < 1$, for some constant $C_1, C_2 > 0$ of $p$ only, we have*

$$\mathbb{P}(a_n^{(3)} \leq e^{-C_1 n}) \geq 1 - e^{-C_2 n},$$

*which implies $\mathbb{E}a_n^{(3)} \leq e^{-C_3 n}$ for some positive constant $C_3$.*

*Proof.* The spirit of this proof is the same as Proposition 1. Instead of studying the change of $M_i$, we can directly focus on $a_i^{(3)}$ when $\theta^* = 0$. Notably, when $\theta^* = 0$, there are only two potential locations of $X_i$.

**Case 1:** $X_i \in I_2$. A correct label ($Y_i = 1$ with probability $1 - p$) leads to $a_i^{(3)} = 2p a_{i-1}^{(3)}$, while a wrong label leads to $a_i^{(3)} = 2(1 - p)a_{i-1}^{(3)}$. As a result,

$$\mathbb{E}\left\{\ln\left(\frac{a_i^{(3)}}{a_{i-1}^{(3)}}\right)\right\} = (1 - p)\ln(2p) + p\ln(2(1 - p)) < 0.$$

**Case 2:** $X_i \in I_3$. Now, a correct label ($Y_i = 0$ with probability $1 - p$) leads to $a_i^{(3)} = 1 - 2(1 - p)a_{i-1}^{(3)}$, while a wrong label leads to $a_i^{(3)} = 1 - 2p a_{i-1}^{(3)}$. Therefore, by Jensen's inequality, we have

$$\mathbb{E}\left\{\ln\left(\frac{a_i^{(3)}}{a_{i-1}^{(3)}}\right)\right\} \leq \ln\left\{\mathbb{E}\left(\frac{a_i^{(3)}}{a_{i-1}^{(3)}}\right)\right\}$$
$$= \ln\left\{(1 - p)\frac{1 - 2(1 - p)a_{i-1}^{(3)}}{1 - a_{i-1}^{(3)}} + p\frac{1 - 2p a_{i-1}^{(3)}}{1 - a_{i-1}^{(3)}}\right\}$$
$$= \ln\left\{1 - \frac{a_{i-1}^{(3)}}{1 - a_{i-1}^{(3)}}(1 - 2p)^2\right\}$$
$$< 0.$$

With a similar argument as Proposition 1, we only have to show that Case 1 occurs sufficiently many times. Specifically, we define $\widetilde{\tau}_i = \inf_{t > \widetilde{\tau}_{i-1}} \text{sign}(a_t^{(3)} \neq \text{sign}(a_{t-1}^{(3)}), i = 1, 2, \ldots, \widetilde{\tau}_0 = 0$, and $\widetilde{T} = \sup_{i \geq 0}\{\widetilde{\tau}_i \leq n\}$. We show $\mathbb{E}\widetilde{T} \geq \eta n$ for some constant $\eta$ by tracking $a_i^{(2)}$. When $X_i \in I_3$, a correct label ($Y_i = 0$ with probability $1-p$) leads to $a_i^{(2)} = 2(1-p)a_{i-1}^{(2)}$, while a wrong label leads to $a_i^{(2)} = 2p a_{i-1}^{(2)}$. Therefore, we have

$$\mathbb{E}\left\{\ln\left(\frac{a_i^{(2)}}{a_{i-1}^{(2)}}\right)\right\} = (1 - p)\ln(2(1 - p)) + p\ln(2p) = \ln(2) - H(p) > 0.$$

The rest of proof is akin to Proposition 2. We thus complete the proof. □

## A.2. Proof of Theorem 2.

*Proof.* **Estimator: constructing $\widehat{g}_n(\cdot)$ by grid–lines–interpolate.**

Pick an integer $M \geq 2$ and set $h = 1/M$. For each multi-index $\widetilde{\ell} \in \{0, \ldots, M\}^{d-1}$ let the base-grid node be $\widetilde{\boldsymbol{x}}_{\widetilde{\ell}} := M^{-1}\widetilde{\ell} \in [0,1]^{d-1}$. Along the vertical line $L_{\widetilde{\ell}} = \{(\widetilde{\boldsymbol{x}}_{\widetilde{\ell}}, x_d) : x_d \in [0,1]\}$, we collect $N$ samples and run a 1-d threshold estimator (using PBA as described in Section 2) to obtain $\widehat{g}(\widetilde{\boldsymbol{x}}_{\widetilde{\ell}})$ as an estimate $g^*(\widetilde{\boldsymbol{x}}_{\widetilde{\ell}})$. This yields a total of $N(M+1)^{d-1}$ samples, where the total number of samples $n$ satisfying $n \geq N(M+1)^{d-1}$. We then interpolate the estimates of $g^*$ at these points to construct a final estimates of the decision boundary.

In particular, we begin by dividing $[0,1]^{d-1}$ in to cells. Without of generality, we assume that $\alpha > 1$ ($\alpha = 1$ can be handled in similar way) and $\frac{M}{\lfloor \alpha \rfloor}$ is an integer (since this can always be achieved by the proper choice of $M$). For the ease of notation, let $r := \lfloor \alpha \rfloor \in \{1, 2, \ldots\}$, and let the cell index $\widetilde{q} = (\widetilde{q}_1, \ldots, \widetilde{q}_{d-1}) \in \{0, \ldots, \frac{M}{r} - 1\}^{d-1}$ define the axis-aligned cell $I_{\widetilde{q}} = \prod_{i=1}^{d-1}[\frac{r\widetilde{q}_i}{M}, \frac{r(\widetilde{q}_i+1)}{M}]$. In this way, the $(r+1)^{d-1}$ lattice nodes inside $I_{\widetilde{q}}$ have multi-indices $\widetilde{\ell} = (\ell_1, \ldots, \ell_{d-1}), \ell_i \in \{r\widetilde{q}_i, r\widetilde{q}_i + 1, \ldots, r\widetilde{q}_i + r\}$, and coordinates $\widetilde{\boldsymbol{x}}_{\widetilde{\ell}} := M^{-1}\widetilde{\ell}$. For bookkeeping in coordinate $i$, set the node locations $z_{i,j} := \frac{r\widetilde{q}_i+j}{M}, j = 0, 1, \ldots, r$, and the local index of $\ell_i$ within its cell $m_i := \ell_i - r\widetilde{q}_i \in \{0, 1, \ldots, r\}$.

Given these notations, we construct $\widehat{g}_n(\cdot)$ by the piecewise polynomial, shown as follows.

$$\widehat{g}_n(\widetilde{\boldsymbol{x}}) = \sum_{\widetilde{q}} \widehat{L}_{\widetilde{q}}(\widetilde{\boldsymbol{x}}) \, \mathbf{1}\{\widetilde{\boldsymbol{x}} \in I_{\widetilde{q}}\}, \tag{10}$$

where $\widehat{L}_{\widetilde{q}}(\widetilde{\boldsymbol{x}}) = \sum_{\widetilde{\ell}: \widetilde{\boldsymbol{x}}_{\widetilde{\ell}} \in I_{\widetilde{q}}} \widehat{g}(\widetilde{\boldsymbol{x}}_{\widetilde{\ell}}) \, Q_{\widetilde{q},\widetilde{\ell}}(\widetilde{\boldsymbol{x}})$, and $Q_{\widetilde{q},\widetilde{\ell}}(\widetilde{\boldsymbol{x}})$ is the multidimensional tensor-product basis on the cell. In particular,

$$Q_{\widetilde{q},\widetilde{\ell}}(\widetilde{\boldsymbol{x}}) := \prod_{i=1}^{d-1} L_{i,\widetilde{q}_i,\ell_i}(\widetilde{\boldsymbol{x}}_i) = \prod_{i=1}^{d-1} \prod_{\substack{j=0 \\ j \neq m_i}}^{r} \frac{\widetilde{\boldsymbol{x}}_i - \frac{r\widetilde{q}_i+j}{M}}{\frac{\ell_i}{M} - \frac{r\widetilde{q}_i+j}{M}},$$

where $L_{i,\widetilde{q}_i,\ell_i}(t) := \prod_{\substack{j=0 \\ j \neq m_i}}^{r} \frac{t - z_{i,j}}{z_{i,m_i} - z_{i,j}}$. $\widehat{g}_n(\cdot)$ defines a classification rule $\widehat{G}_n$.

By Equation 10, we have the follows.

$$\mathcal{O}(\|\widehat{g}_n - g^*\|_1) = \mathcal{O}\Big(\sum_{\widetilde{q}} \|(\widehat{L}_{\widetilde{q}} - g^*)\mathbf{1}\{\widetilde{\boldsymbol{x}} \in I_{\widetilde{q}}\}\|_{L^1([0,1]^{d-1})}\Big)$$

$$= \mathcal{O}\Big(\sum_{\widetilde{q}} \|(L_{\widetilde{q}} - g^*)\mathbf{1}\{\widetilde{\boldsymbol{x}} \in I_{\widetilde{q}}\} + (\widehat{L}_{\widetilde{q}} - L_{\widetilde{q}})\mathbf{1}\{\widetilde{\boldsymbol{x}} \in I_{\widetilde{q}}\}\|_{L^1([0,1]^{d-1})}\Big)$$

$$= \mathcal{O}\Big(\sum_{\widetilde{q}} \|(L_{\widetilde{q}} - g^*)\mathbf{1}\{\widetilde{\boldsymbol{x}} \in I_{\widetilde{q}}\}\|_{L^1([0,1]^{d-1})} + \|(\widehat{L}_{\widetilde{q}} - L_{\widetilde{q}})\mathbf{1}\{\widetilde{\boldsymbol{x}} \in I_{\widetilde{q}}\}\|_{L^1([0,1]^{d-1})}\Big),$$

where $L_{\widetilde{q}}(\widetilde{\boldsymbol{x}}) = \sum_{\widetilde{\ell}: \widetilde{\boldsymbol{x}}_{\widetilde{\ell}} \in I_{\widetilde{q}}} g^*(\widetilde{\boldsymbol{x}}_{\widetilde{\ell}}) \, Q_{\widetilde{q},\widetilde{\ell}}(\widetilde{\boldsymbol{x}})$ is the Clairvoyant version of $\widehat{L}_{\widetilde{q}}$.

Note that

$$\|(L_{\widetilde{q}} - g^*)\mathbf{1}\{\widetilde{\boldsymbol{x}} \in I_{\widetilde{q}}\}\|_{L^1([0,1]^{d-1})} = \int_{I_{\widetilde{q}}} |L_{\widetilde{q}}(\widetilde{\boldsymbol{x}}) - g^*(\widetilde{\boldsymbol{x}})|d\widetilde{\boldsymbol{x}} = \mathcal{O}\Big(\int_{I_{\widetilde{q}}} M^{-\alpha}d\widetilde{\boldsymbol{x}}\Big), \tag{11}$$

by using Lemma 3 and resulting in $\mathcal{O}(M^{-\alpha}M^{-(d-1)})$. Moreover, by conditioning on the good event where $|\widehat{g}(\widetilde{\boldsymbol{x}}_{\widetilde{\ell}}) -$

$g^*(\widetilde{\boldsymbol{x}}_{\widetilde{l}})| \leq \epsilon_N$, we have

$$\|(\widehat{L}_{\widetilde{q}} - L_{\widetilde{q}})\mathbf{1}\{\widetilde{\boldsymbol{x}} \in I_{\widetilde{q}}\}\|_{L^1([0,1]^{d-1})} = \sum_{\widetilde{l}:\widetilde{\boldsymbol{x}}_{\widetilde{l}} \in I_{\widetilde{q}}} |\widehat{g}(\widetilde{\boldsymbol{x}}_{\widetilde{l}}) - g^*(\widetilde{\boldsymbol{x}}_{\widetilde{l}})|\|Q_{\widetilde{q},\widetilde{l}}\|_{L^1([0,1]^{d-1})} \tag{12}$$

$$\leq \sum_{\widetilde{l}:\widetilde{\boldsymbol{x}}_{\widetilde{l}} \in I_{\widetilde{q}}} \epsilon_N \left( \int_{I_{\widetilde{q}}} Q_{\widetilde{q},\widetilde{l}}(\widetilde{x}) d\mu\widetilde{\boldsymbol{x}} \right) \tag{13}$$

$$\leq \sum_{\widetilde{l}:\widetilde{\boldsymbol{x}}_{\widetilde{l}} \in I_{\widetilde{q}}} \epsilon_N \left( \int_{I_{\widetilde{q}}} r^{(d-1)r} d\mu\widetilde{\boldsymbol{x}} \right) \tag{14}$$

$$= \mathcal{O}\left( \epsilon_N M^{-(d-1)} \right). \tag{15}$$

Note that $\mu$ is a Lebesgue measure of $\widetilde{\boldsymbol{x}}$ which is uniform on $[0,1]^{d-1}$. By Equation 11 and 12, we have

$$\|\widehat{g}_n - g^*\|_1 \leq \mathcal{O}\Big( \sum_{\widetilde{q}} \|(L_{\widetilde{q}} - g^*)\mathbf{1}\{\widetilde{\boldsymbol{x}} \in I_{\widetilde{q}}\}\|_{L^1([0,1]^{d-1})} + \|(\widehat{L}_{\widetilde{q}} - L_{\widetilde{q}})\mathbf{1}\{\widetilde{\boldsymbol{x}} \in I_{\widetilde{q}}\}\|_{L^1([0,1]^{d-1})} \Big)$$

$$\leq \mathcal{O}\left( M^{d-1}(M^{-\alpha}M^{-(d-1)} + \epsilon_N M^{-(d-1)}) \right)$$

$$= \mathcal{O}\left( M^{-\alpha} + \epsilon_N \right).$$

According to Theorem 1, we know $\mathbb{P}(|\widehat{g}(\widetilde{\boldsymbol{x}}_{\widetilde{l}}) - g^*(\widetilde{\boldsymbol{x}}_{\widetilde{l}})| > \epsilon_N) \leq \frac{3}{\epsilon_N} \exp(-CN)$. Therefore, we choose $N = \lceil K \log n \rceil$, where $K > \frac{2\alpha}{C(d-1)}$, $M = \lfloor \left( \frac{n}{K \log n} \right)^{1/(d-1)} \rfloor$ and $\epsilon_N = \sqrt{3}\,e^{-cN/2}$, leading to

$$\mathbb{E}\|\widehat{g}_n - g^*\|_1 \leq \mathcal{O}\left( M^{-\alpha} + \epsilon_N \right) + \frac{3}{\epsilon_N} \exp(-CN) = \mathcal{O}\left( \left( \frac{\log n}{n} \right)^{\frac{\alpha}{d-1}} \right).$$

$\square$

**Lemma 3.** $\sup_{g^* \in \Sigma(L,\alpha)} \max_{\widetilde{\boldsymbol{x}} \in I_{\widetilde{q}}} |L_{\widetilde{q}}(\widetilde{\boldsymbol{x}}) - g^*(\widetilde{\boldsymbol{x}})| = \mathcal{O}(M^{-\alpha})$.

*Proof.* Let $\widetilde{\boldsymbol{x}} \in I_{\widetilde{q}}$ and $g \in \Sigma(L,\alpha)$, we have the follows.

$$\left| L_{\widetilde{q}}(\widetilde{\boldsymbol{x}}) - g^*(\widetilde{\boldsymbol{x}}) \right| = \left| L_{\widetilde{q}}(\widetilde{\boldsymbol{x}}) - \mathrm{TP}_{\widetilde{q}rM^{-1}}(\widetilde{\boldsymbol{x}}) - g^*(\widetilde{\boldsymbol{x}}) + \mathrm{TP}_{\widetilde{q}rM^{-1}}(\widetilde{\boldsymbol{x}}) \right|$$

$$\leq \left| L_{\widetilde{q}}(\widetilde{\boldsymbol{x}}) - \mathrm{TP}_{\widetilde{q}rM^{-1}}(\widetilde{\boldsymbol{x}}) \right| + \left| g^*(\widetilde{\boldsymbol{x}}) - \mathrm{TP}_{\widetilde{q}rM^{-1}}(\widetilde{\boldsymbol{x}}) \right|$$

$$\leq \left| L_{\widetilde{q}}(\widetilde{\boldsymbol{x}}) - \mathrm{TP}_{\widetilde{q}rM^{-1}}(\widetilde{\boldsymbol{x}}) \right| + L\left\| \widetilde{\boldsymbol{x}} - \widetilde{q}rM^{-1} \right\|^{\alpha}$$

$$\leq \left| L_{\widetilde{q}}(\widetilde{\boldsymbol{x}}) - \mathrm{TP}_{\widetilde{q}rM^{-1}}(\widetilde{\boldsymbol{x}}) \right| + \mathcal{O}(M^{-\alpha}).$$

Note that the tensor–polynomial approximation space contains the space of degree $r$ polynomials. Therefore we can write

$L_{\widetilde{q}}(\widetilde{\boldsymbol{x}})$ as a tensor–product polynomial. Therefore, we have

$$
\begin{aligned}
\left| L_{\widetilde{q}}(\widetilde{\boldsymbol{x}}) - g^*(\widetilde{\boldsymbol{x}}) \right| &\leq \left| \sum_{\widetilde{l}:\boldsymbol{x}_{\widetilde{l}} \in I_{\widetilde{q}}} g^*(\widetilde{\boldsymbol{x}}_{\widetilde{l}}) \, Q_{\widetilde{q},\widetilde{l}}(\widetilde{\boldsymbol{x}}) - \mathrm{TP}_{\widetilde{q}rM^{-1}}(\widetilde{\boldsymbol{x}}) \right| + \mathcal{O}(M^{-\alpha}) \\
&= \left| \sum_{\widetilde{l}:\boldsymbol{x}_{\widetilde{l}} \in I_{\widetilde{q}}} \left( g^*(\widetilde{\boldsymbol{x}}_{\widetilde{l}}) - \mathrm{TP}_{\widetilde{q}rM^{-1}}(\widetilde{\boldsymbol{x}}_{\widetilde{l}}) \right) Q_{\widetilde{q},\widetilde{l}}(\widetilde{\boldsymbol{x}}) \right| + \mathcal{O}(M^{-\alpha}) \\
&\leq \sum_{\widetilde{l}:\boldsymbol{x}_{\widetilde{l}} \in I_{\widetilde{q}}} \left| g^*(\widetilde{\boldsymbol{x}}_{\widetilde{l}}) - \mathrm{TP}_{\widetilde{q}rM^{-1}}(\widetilde{\boldsymbol{x}}_{\widetilde{l}}) \right| \left| Q_{\widetilde{q},\widetilde{l}}(\widetilde{\boldsymbol{x}}) \right| + \mathcal{O}(M^{-\alpha}) \\
&\leq \sum_{\widetilde{l}:\boldsymbol{x}_{\widetilde{l}} \in I_{\widetilde{q}}} L \left\| \widetilde{\boldsymbol{x}} - \widetilde{q}rM^{-1} \right\|^{\alpha} \left| Q_{\widetilde{q},\widetilde{l}}(\widetilde{\boldsymbol{x}}) \right| + \mathcal{O}(M^{-\alpha}) \\
&\leq \sum_{\widetilde{l}:\boldsymbol{x}_{\widetilde{l}} \in I_{\widetilde{q}}} L \left\| \widetilde{\boldsymbol{x}} - \widetilde{q}rM^{-1} \right\|^{\alpha} r^{(d-1)r} + \mathcal{O}(M^{-\alpha}) \\
&\leq \sum_{\widetilde{l}:\boldsymbol{x}_{\widetilde{l}} \in I_{\widetilde{q}}} \mathcal{O}(M^{-\alpha}) + \mathcal{O}(M^{-\alpha}) = r^{d-1} \mathcal{O}(M^{-\alpha}) + \mathcal{O}(M^{-\alpha}) = \mathcal{O}(M^{-\alpha}).
\end{aligned}
$$

$\square$

## A.3. Proof of Theorem 3

*Proof.* When the decision boundary is linear, the proof proceeds in two stages. In $d$ dimensions, a linear decision boundary induces a hyperplane that intersects a subset of the $d2^{d-1}$ edges of the unit cube $[0,1]^d$. Each such intersection is characterized by a change in the label distribution along the corresponding edge.

In the first step, we perform edge detection via hypothesis testing. For each edge of the cube, we query the two endpoints and test whether the conditional label distributions differ across the edge. Using a two-sample test and a union bound over all $d2^{d-1}$ edges, we show that, with probability at least $1 - C \exp(-cn)$, with some positive constant $C$ and $c$, the procedure correctly identifies the entire set of edges intersected by the decision boundary, while making no false discoveries.

In the second step, for each detected edge, we localize the intersection point by running PBA along the edge to estimate the threshold at which the label switches. Finally, we reconstruct the hyperplane parameters by solving a linear regression problem using these estimated intersection points. Combining the exponential localization error of PBA with the stability of the linear reconstruction yields an overall exponential convergence rate for the estimated decision boundary.

**Step 1: Edge Detection via Hypothesis Testing.** We begin by identifying the set of cube edges intersected by the linear decision boundary. Let $\mathcal{V} := \{0,1\}^d$ denote the set of vertices of the unit cube $[0,1]^d$, and let

$$
\mathcal{E} := \big\{ \{u,v\} \subset \mathcal{V} : \|u - v\|_0 = 1 \big\}
$$

be the set of all edges, where $\|\cdot\|_0$ denotes the Hamming distance. The total number of edges is $|\mathcal{E}| = d \, 2^{d-1}$. For each edge $E = \{u,v\} \in \mathcal{E}$, we query the classifier at its two endpoints. Let $\mu_{E,0} := \mathbb{P}(Y = 1 \mid X = u)$, $\mu_{E,1} := \mathbb{P}(Y = 1 \mid X = v)$, and $\delta_E := \mu_{E,1} - \mu_{E,0}$. If the decision boundary intersects the edge $E$, then $\mu_{E,0}$ and $\mu_{E,1}$ differ, and hence $\delta_E \neq 0$. Conversely, if the boundary does not intersect $E$, then $\mu_{E,0} = \mu_{E,1}$ and $\delta_E = 0$. Define the (unknown) set of cut edges

$$
\mathcal{C} := \{E = \{u,v\} \in \mathcal{E} : \ \delta_E \neq 0\}.
$$

For each edge $E = \{u,v\}$, we draw $n$ i.i.d. labels at each endpoint and form

$$
\widehat{\mu}_{E,0} := \frac{1}{n} \sum_{k=1}^{n} Y_{E,k}^{(0)}, \qquad \widehat{\mu}_{E,1} := \frac{1}{n} \sum_{k=1}^{n} Y_{E,k}^{(1)}, \qquad \widehat{\delta}_E := \widehat{\mu}_{E,1} - \widehat{\mu}_{E,0}.
$$

We construct the set of cut edges $\widehat{\mathcal{C}} := \{E \in \mathcal{E} : |\widehat{\delta}_E| \geq \tau_{n,d}\}$, for a threshold $\tau_{n,d} > 0$ specified below to perform PBA in the second stage. In the following, we bound the probability of any edge-identification error by

$$
\begin{aligned}
\mathbb{P}(\widehat{\mathcal{C}} \neq \mathcal{C}) &= \mathbb{P}\Big(\exists\, E \in \mathcal{E} : \mathbf{1}\{E \in \widehat{\mathcal{C}}\} \neq \mathbf{1}\{E \in \mathcal{C}\}\Big) \\
&\leq \mathbb{P}\Big(\exists\, E \in \mathcal{E} \setminus \mathcal{C} : E \in \widehat{\mathcal{C}}\Big) + \mathbb{P}\Big(\exists\, E \in \mathcal{C} : E \notin \widehat{\mathcal{C}}\Big) \\
&\leq \sum_{E \in \mathcal{E} \setminus \mathcal{C}} \mathbb{P}(|\widehat{\delta}_E| \geq \tau_{n,d}) + \sum_{E \in \mathcal{C}} \mathbb{P}(|\widehat{\delta}_E| < \tau_{n,d}).
\end{aligned}
$$

Let $D_E := \widehat{\delta}_E - \delta_E = (\widehat{\mu}_{E,1} - \mu_{E,1}) - (\widehat{\mu}_{E,0} - \mu_{E,0})$. By Hoeffding's inequality and a union bound over endpoints, for any $t > 0$,

$$
\begin{aligned}
\mathbb{P}(|D_E| \geq t) &\leq \mathbb{P}\big(|\widehat{\mu}_{E,1} - \mu_{E,1}| \geq t/2\big) + \mathbb{P}\big(|\widehat{\mu}_{E,0} - \mu_{E,0}| \geq t/2\big) \\
&\leq 2\exp\Big(-2n(t/2)^2\Big) + 2\exp\Big(-2n(t/2)^2\Big) = 4\exp\Big(-\frac{nt^2}{2}\Big).
\end{aligned}
\tag{16}
$$

If $E \notin \mathcal{C}$ then $\delta_E = 0$ and $\widehat{\delta}_E = D_E$, hence

$$
\mathbb{P}(E \in \widehat{\mathcal{C}} \mid E \notin \mathcal{C}) = \mathbb{P}(|\widehat{\delta}_E| \geq \tau_{n,d} \mid \delta_E = 0) \leq 4\exp\Big(-\frac{n\tau_{n,d}^2}{2}\Big).
\tag{17}
$$

Since for all $E \in \mathcal{C}$, $|\delta_E| = 1 - 2p$. Denote $\Delta = 1 - 2p$, for any $\tau_{n,d} \in (0, \Delta)$, we have

$$
\{|\widehat{\delta}_E| < \tau_{n,d}\} \subseteq \{|\widehat{\delta}_E - \delta_E| \geq |\delta_E| - \tau_{n,d}\} \subseteq \{|D_E| \geq \Delta - \tau_{n,d}\},
$$

so by Eqn. (16), we have

$$
\mathbb{P}(|\widehat{\delta}_E| < \tau_{n,d} \mid \delta_E = \Delta) \leq P(|D_E| \geq \Delta - \tau_{n,d}) \leq 4\exp\Big(-\frac{n(\Delta - \tau_{n,d})^2}{2}\Big).
\tag{18}
$$

Combining (17)–(18) and using $|\mathcal{C}| \leq |\mathcal{E}|$ gives

$$
\mathbb{P}(\widehat{\mathcal{C}} \neq \mathcal{C}) \leq 4|\mathcal{E}|\exp\Big(-\frac{n\tau_{n,d}^2}{2}\Big) + 4|\mathcal{E}|\exp\Big(-\frac{n(\Delta - \tau_{n,d})^2}{2}\Big).
\tag{19}
$$

Let $M := |\mathcal{E}| = d\,2^{d-1}$ and choose

$$
\tau_{n,d} := \frac{\Delta}{2} - \sqrt{\frac{\log(8M)}{n}},
\tag{20}
$$

where we require $n > \frac{4\log(8M)}{\Delta^2}$ so that $\tau_{n,d} > 0$. Then $\Delta - \tau_{n,d} = \frac{\Delta}{2} + \sqrt{\frac{\log(8M)}{n}}$ and

$$
\exp\Big(-\frac{n(\Delta - \tau_{n,d})^2}{2}\Big) \leq \exp\Big(-\frac{n\tau_{n,d}^2}{2}\Big)\exp\big(-\log(8M)\big),
$$

so the second term in (19) is dominated by the first, and we obtain

$$
\mathbb{P}(\widehat{\mathcal{C}} \neq \mathcal{C}) \leq 8M\exp\Big(-\frac{n\tau_{n,d}^2}{2}\Big) \leq \exp(-cn) \quad \text{for all sufficiently large } n,
\tag{21}
$$

for some constant $c > 0$ depending only on $\Delta$ and $d$. With a slight abuse of notation, we now let $n$ denote the total number of queried samples. Consequently, the number of samples collected at each vertex is $n/2^d$. We require $\frac{n}{2^d} \geq \frac{4\log M}{\Delta^2} \asymp O\left(\frac{d\log d}{(1-p)^2}\right)$, under which the probability of incorrect edge identification satisfies

$$
\mathbb{P}(\widehat{\mathcal{C}} \neq \mathcal{C}) \leq C\exp\left(-c\frac{n}{2^d}\right)
$$

for some constants $C, c > 0$ depending only on $p$ and $d$. In the following, we show that, conditioning on the good event $\widehat{\mathcal{C}} = \mathcal{C}$, running PBA achieves exponential convergence rate.

**Step 2: Running PBA on $\widehat{\mathcal{C}}$.** Without loss of generality, we assume that $\widehat{\mathcal{C}}$ consists of edges perpendicular to the last dimension $x_d$; otherwise, we can rotate the coordinate system. We thus denote $g^*(\widetilde{\boldsymbol{x}}) = a_*^\top \widetilde{\boldsymbol{x}} + b_*, \widetilde{\boldsymbol{x}} \in [0, 1]^{d-1}$.

We first pick $m \geq d$ anchor points $\widetilde{\boldsymbol{x}}_1, \ldots, \widetilde{\boldsymbol{x}}_m$, where $(\widetilde{\boldsymbol{x}}_i, 1) \in \widehat{\mathcal{C}}$ and the augmented design

$$Z = \begin{bmatrix} \widetilde{\boldsymbol{x}}_1^\top & 1 \\ \vdots & \vdots \\ \widetilde{\boldsymbol{x}}_m^\top & 1 \end{bmatrix} \in \mathbb{R}^{m \times d}$$

satisfies $\mathrm{rank}(Z) = d$. For each fixed $\widetilde{\boldsymbol{x}}_j$, query along the vertical line $\{(\widetilde{\boldsymbol{x}}_j, t) : t \in [0, 1]\}$ and run a PBA to estimate the one-dimensional threshold $t_j^* := g^*(\widetilde{\boldsymbol{x}}_j) = a_*^\top \widetilde{\boldsymbol{x}}_j + b_*$.

Let $\widehat{t}_j$ be the PBA estimate after $n_j$ queries on line $j$, and set $\widehat{t} = (\widehat{t}_1, \ldots, \widehat{t}_d)^\top$. Estimate $\theta_* := (a_*, b_*) \in \mathbb{R}^d$ by least squares: $\widehat{\theta}_n := \arg\min_{\theta \in \mathbb{R}^d} \|\widehat{t} - Z\theta\|_2^2 = (Z^\top Z)^{-1} Z^\top \widehat{t}$, where $\widehat{t} = \widehat{g}(\widetilde{\boldsymbol{x}}) := \widehat{a}^\top \widetilde{\boldsymbol{x}} + \widehat{b}$. Let the total sample be $n = \sum_{j=1}^d n_j$. Let $\varepsilon := (\varepsilon_1, \ldots, \varepsilon_d)^\top$ with $\varepsilon_j = \widehat{t}_j - t_j^*$. Then we can express

$$\widehat{\theta}_n - \theta_* = (Z^\top Z)^{-1} Z^\top \varepsilon,$$

and $\|\widehat{\theta}_n - \theta_*\|_2 \leq \frac{1}{\sigma_{\min}(Z)} \|\varepsilon\|_2$, where $\sigma_{\min}(Z) > 0$ is the smallest singular value of $Z$. Let $\Delta a := \widehat{a} - a_*$ and $\Delta b := \widehat{b} - b_*$. Then we have

$$\mathbb{E}\|\widehat{g}_n - g^*\|_1 := \mathbb{E}\int_{[0,1]^{d-1}} |\Delta a^\top u + \Delta b| \, du \leq |\Delta b| + \frac{1}{2}\sum_{k=1}^{d-1} |\Delta a_k| \leq \left(1 + \frac{\sqrt{d-1}}{2}\right)\|\widehat{\theta} - \theta_*\|_2$$

$$\leq \frac{\left(1 + \frac{\sqrt{d-1}}{2}\right)}{\sigma_{\min}(Z)} \mathbb{E}\|\varepsilon\|_2 = \frac{\left(1 + \frac{\sqrt{d-1}}{2}\right)}{\sigma_{\min}(Z)} \mathbb{E}\|\widehat{t} - t^*\|_2 \leq \frac{\left(1 + \frac{\sqrt{d-1}}{2}\right)}{\sigma_{\min}(Z)} \sqrt{\sum_{j=1}^d \left(\mathbb{E}|\varepsilon_j|\right)^2}$$

$$\leq 3\frac{\left(1 + \frac{\sqrt{d-1}}{2}\right)}{\sigma_{\min}(Z)} \sqrt{d} \max_j \exp(-Cn_j).$$

We have the last equation by Theorem 1, which shows $\mathbb{E}|\widehat{t}_j - t_j^*| \leq 3\exp(-Cn_j), \forall j$, where $C$ is a constant of $p$ only. By taking $n_j = \frac{n}{d}$, we show that $\mathbb{E}\|\widehat{g} - g^*\|_1 \leq 3\frac{\left(1 + \frac{\sqrt{d-1}}{2}\right)}{\sigma_{\min}(Z)} \sqrt{d}\exp(-C\frac{n}{d})$. Since the matrix $Z$ has full column rank and $\sigma_{\min}(Z)$ is bounded below by a positive constant depending only on $d$, we can write $\mathbb{E}\|\widehat{g} - g^*\|_1 \leq C_1 \exp(-cn)$, where $C_1 > 0$ depends only on $d$ and $c > 0$ depends only on $p$ and $d$. $\qquad\square$

## B. Discussion on the Noise Level

Our results apply to general responses $Y$ with a noise level up to $p$. That is, $\mathbb{P}(Y = h_{\theta^*}(X)) \geq 1 - p$ and $\mathbb{P}(Y = 1 - h_{\theta^*}(X)) \leq p$. To see it, an intuitive explanation is that at each round $i$, a correct response will drive the PBA estimator to be closer to the truth $\theta^*$, while an incorrect response will push it away from the truth. As a result, a higher noise level corresponds to a harder learning problem, and we discuss the most difficult learning scenario ($\mathbb{P}(Y = 1 - h_{\theta^*}(X)) = p$) in the main paper.

Technically, inspecting the proof of Proposition 1, we find that the expectation of $M_i/M_{i-1}$ is even smaller when the probability of incorrect label is smaller than $p$. Meanwhile, the crossing time $T$ is still guaranteed to be at the order of $O(n)$. Therefore, the probability of an ill-performed estimator remains an exponentially decaying rate.

## C. Pseudo-code for high-dimensional PBA

In this section, we present the pseudo-code for the estimators used in the proofs of Theorems 2 and 3, given in Algorithm 1 and Algorithm 2, respectively.

---

**Algorithm 1** PBA for Hölder smooth classifier

---

**Require:** The query size $n$, dimension $d$, smoothness $\alpha$, and the noise level $p$

1: Choose $N = \lceil K \log n \rceil$, where $K > \frac{2\alpha}{C(d-1)}$, $M = \lfloor \left(\frac{n}{K \log n}\right)^{1/(d-1)} \rfloor$ and $C$ is a constant of $p$ in Theorem 1
2: Divide $[0,1]^{d-1}$ to $M^{d-1}$ equally-spaced points $\widetilde{\boldsymbol{x}}_{\widetilde{\ell}}$.
3: Apply one-dimensional PBA with $N$ queries along each vertical line $L_{\widetilde{\ell}} = \{(\widetilde{\boldsymbol{x}}_{\widetilde{\ell}}, x_d) : x_d \in [0,1]\}$
4: Let $\widehat{g}_n$ be the interpolated estimator given in Eq. 10.

**Ensure:** The estimated boundary $\widehat{g}_n$

---

**Algorithm 2** PBA for linear classifier

---

**Require:** The query size $n$, dimension $d$, and the noise level $p$

1: Query each vertex of the hypercube $[0,1]^{d-1}$ for $n/2^{d+1}$ times.
2: Run a hypothesis testing on each edge of the hypercube, with null hypothesis being the vertices of this edge has the same label, forming a set of cut edges $\widehat{\mathcal{C}}$.
3: **if** $\widehat{\mathcal{C}}$ cannot be the intersection of a linear boundary and hypercube **then**
4:     Return $\widehat{g}_n = 0$                                                                                 ▷ Failed
5: **end if**
6: Find $d$ anchor edges in $\widehat{\mathcal{C}}$ that are full-rank
7: **for** Each anchor edge **do**
8:     Apply one-dimensional PBA with $n/(d2^{d+1})$ queries to find the approximation of intersection point
9: **end for**
10: $\widehat{g}_n \leftarrow$ Interpolation of the intersection points identified above.

**Ensure:** The estimated boundary $\widehat{g}_n$

---

